# Deep learning for bias correction of MJO prediction

H. Kim [1] ✉, Y. G. Ham [2], Y. S. Joo[2] & S. W. Son[3]

Producing accurate weather prediction beyond two weeks is an urgent challenge due to its ever-increasing socioeconomic value. The Madden-Julian Oscillation (MJO), a planetary-scale tropical convective system, serves as a primary source of global subseasonal (i.e., targeting three to four weeks) predictability. During the past decades, operational forecasting systems have improved substantially, while the MJO prediction skill has not yet reached its potential predictability, partly due to the systematic errors caused by imperfect numerical models. Here, to improve the MJO prediction skill, we blend the state-of-the-art dynamical forecasts and observations with a Deep Learning bias correction method. With Deep Learning bias correction, multi-model forecast errors in MJO amplitude and phase averaged over four weeks are significantly reduced by about 90% and 77%, respectively. Most models show the greatest improvement for MJO events starting from the Indian Ocean and crossing the Maritime Continent.

[1] School of Marine and Atmospheric Sciences, Stony Brook University, New York, NY, USA. [2] Department of Oceanography, Chonnam National University, Gwangju, South Korea. [3] School of Earth and Environmental Sciences, Seoul National University, Seoul, South Korea. ✉email: hyemi.kim@stonybrook.edu

Accurate prediction beyond the two-week limit of atmospheric predictability is extremely valuable to society and the economy. In particular, reliable forecasts in the subseasonal range (i.e., timescale of 3–4 weeks) provide vital information about hazardous weather threats, such as floods, heat waves, and cold spells, which are extremely important for risk managers, stakeholders, and policymakers. The value of subseasonal forecasts has been recognized by society and the scientific community, and tremendous international efforts towards making reliable subseasonal forecasts are underway[1,2].

One of the primary predictability sources for the 3–4 week forecast window in the global climate system is the Madden-Julian Oscillation (MJO)[3], the dominant mode of tropical subseasonal variability. The MJO is a planetary-scale organized convection-circulation coupled system with a typical period of 30–60 days, characterized by an eastward propagation, especially during boreal winter. As the MJO develops and propagates, anomalous diabatic heating leads to the formation of an anomalous Rossby wave source. This excites the Rossby waves to propagate into the extratropics, modulating the weather events therein[4–7]. For example, studies have shown significant influences of the MJO on tropical cyclones[8,9], extreme temperature and precipitation[10–12], storm tracks[13–15], atmospheric blocking events[16], atmospheric rivers[17–19], tornadoes[20], and weather in the Arctic[21] and Antarctic[22], among many others. Due to its far-reaching global impacts[7] and quasiperiodic nature, the MJO is recognized as one of the leading sources of global climate predictability for the subseasonal timescale that bridges the gap between the traditional weather (i.e., from one day to 2 weeks) and seasonal (i.e., from 2 months to 1 year) forecast ranges.

Recent advances in theoretical understanding, improved numerical models, and international collaborative activities on field campaigns and forecast experiments have promoted advances in MJO forecasting[23–25]. Now, the state-of-the-art dynamical forecast systems are able to predict the MJO up to 3 weeks in advance[23,25], a remarkable improvement since the early 2000s. However, due to errors originating from imperfect numerical models, the MJO prediction skill has not reached its theoretical predictability, which is known to be ~7 weeks[26]. This indicates that there is considerable room for improvement[23,25–28]. One of the greatest challenges in current dynamical forecast systems is the fast damping of the MJO signal as the forecast lead time increases, which results in a rapid decrease of forecast skill[25,29,30]. This systematic damping of the MJO convection signal is particularly apparent when the MJO starts over the Indian Ocean and is expected to propagate through the Maritime Continent and move further into the western Pacific. The frequency of MJO events not crossing the Maritime Continent in forecast models is more than twice as large as it is in observations[30], known as the Maritime Continent prediction barrier[25,29–32]. Given that the MJO prediction alone presents considerable systematic biases, the global weather forecast beyond 2 weeks is an even more daunting task.

Model deficiencies in simulating realistic MJO events are partially due to our poor understanding of the underlying physics. Therefore, more efforts on process-level diagnostics are suggested to further improve MJO simulation and prediction[23]. Concurrently, post-processing of numerical forecasts has been proven to be a powerful tool to improve forecasts when models display systematic biases[33]. A recent study has shown an increase of MJO prediction skill by correcting model biases with a linear statistical model[34]. Deep learning (DL) techniques have been proven to be a powerful tool for identifying weather and climate patterns[35–37], sub-grid scale parameterizations[38,39], weather and climate predictions[40–44], and post-processing of numerical weather forecasts (shorter than 7 days[43,44]). However, post-processing with DL methods has not yet been applied to MJO forecasts.

In this study, we utilize DL as a bias correction method to improve MJO forecasts. We demonstrate that the DL post-processing substantially reduces the MJO forecast errors from the state-of-the-art dynamical forecasting systems, thus making strides towards improving global extended range forecasts.

## Results

**Improved MJO prediction with deep learning bias correction.** Figure 1 highlights the advantage of Deep learning bias correction (DL-correction) for MJO forecasts. It shows the multi-model mean of predicted Real-time Multivariate MJO indices (RMMs) composite on a phase-space diagram[45] as a function of initial MJO phases and forecast lead days from day 1 to day 28 (4 weeks). Predictions from the original Subseasonal-to-seasonal (S2S) reforecasts and DL-correction for each forecast target years are composited and compared with observations (see "Methods" section). The composite results of individual models are displayed in Supplementary Fig. 2. Several key results strongly demonstrate the benefit of the DL-correction on MJO forecasts throughout all MJO phases. A large discrepancy between S2S reforecasts and observations is clearly shown on day 1. Most S2S models forecast either weaker (phases 2 and 3) or stronger amplitude (phases 6 and 7), or phase (θ) shifts relative to the observations on day 1. The DL-correction reduces those systematic errors, making the day 1 and the following forecasts closer to the observations in all models and throughout all MJO phases (Fig. 1 and Supplementary Fig. 2).

To evaluate the forecast errors quantitatively, the bivariate root-mean-squared error (BMSE, see "Methods" section) is calculated as a function of initial MJO phases from the

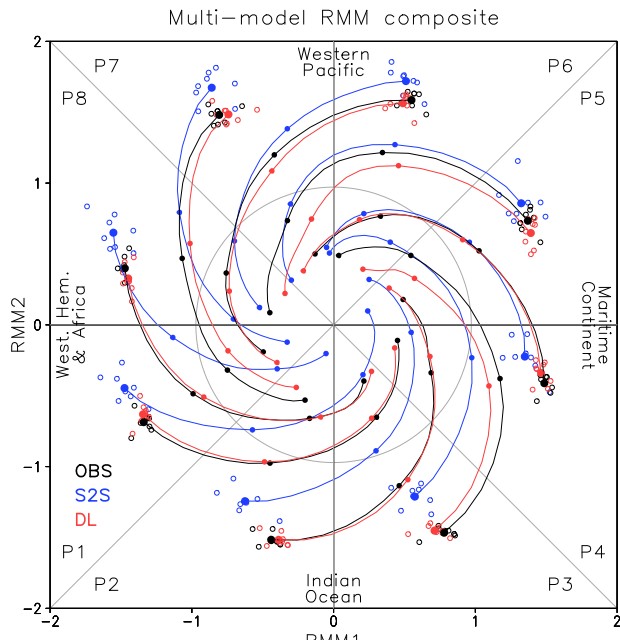

**Fig. 1 Observed and predicted Madden-Julian oscillation (MJO) composites.** Multi-model means of the Real-time Multivariate MJO indices (RMMs) composite on the phase-space diagram in eight MJO phases for observations (black), Subseasonal-to-seasonal (S2S) reforecasts (blue), and Deep learning (DL) corrections (red). Forecasts on day 1 by the eight individual models are depicted as open circles and multi-model means as large closed circles. Small closed circles represent seven-day intervals from day 1. A three-day moving average is applied. Note that Fig. 1 is an average of individual model composite shown in Supplementary Fig. 2.

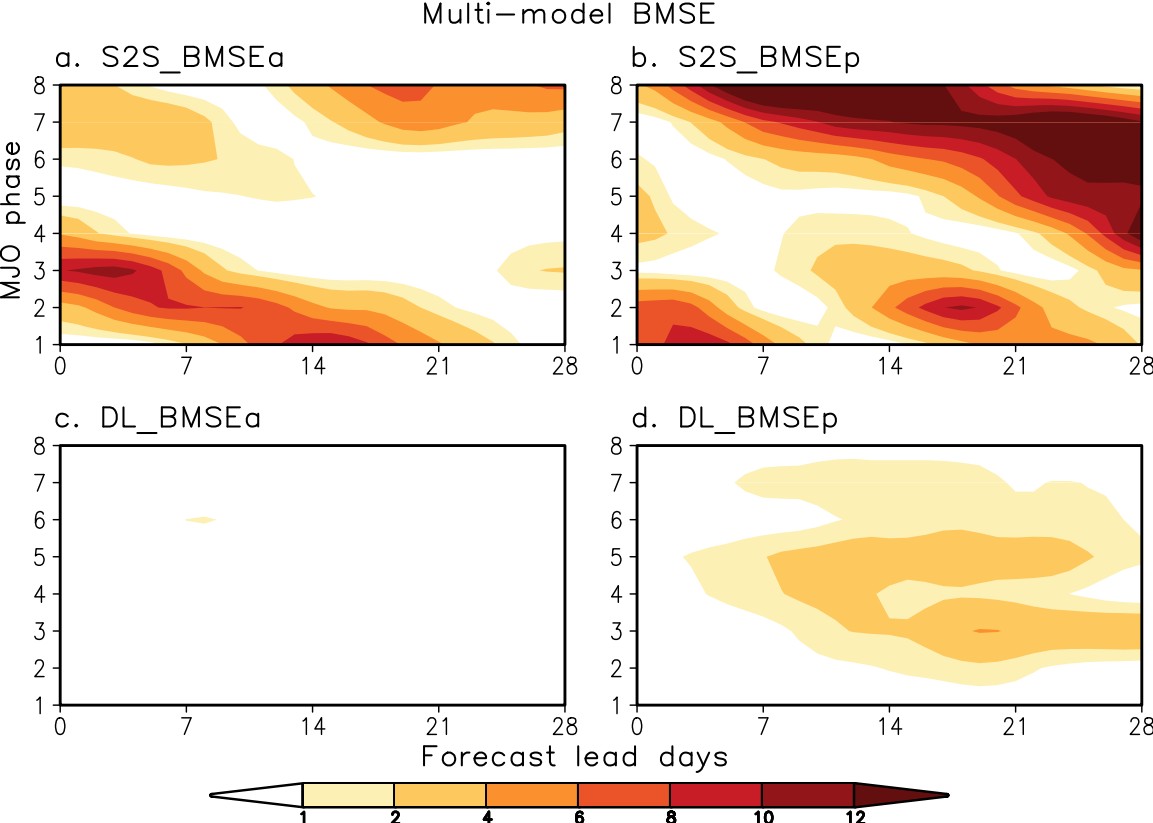

**Fig. 2 Forecast errors in multi-model mean S2S reforecasts and DL-corrections.** Multi-model mean bivariate root-mean-squared error (BMSE) as a function of initial MJO phases and forecast lead days for (top) Subseasonal-to-seasonal (S2S) reforecasts and (bottom) Deep learning (DL)-corrections. BMSE is divided into (**a**, **c**) amplitude error (BMSEa) and (**b**, **d**) phase error (BMSEp) for the Real-time Multivariate MJO indices (RMMs) composite shown in Fig. 1. Values are multiplied by 100. Note that a three-day moving average is applied.

composites shown in Fig. 1. The amplitude error (BMSEa) appears in the S2S models from the beginning of the forecasts for most phases, with predominant errors in phases 2 and 3 (Fig. 2a). This amplitude error reduces substantially when DL-correction is applied (Fig. 2c). Whether amplitude errors are large or small in individual S2S models, they all become similar after the DL-correction (Supplementary Fig. 3). For example, during the first 2 weeks, the ECMWF-Cy43r3 has the largest MJO amplitude error in phases 2 and 3, while the NASA-GEOS5 possesses a large amplitude error in phases 6 and 7 (Supplementary Figs. 2 and 3). Nevertheless, these amplitude errors become negligible after DL-correction (Supplementary Fig. 3). The BMSEa averaged over 4 weeks and eight phases is summarized in Fig. 3. After DL-correction, BMSEa in S2S reforecasts is about 90% reduced in multi-model mean and about 70–94% reduced in individual models (Fig. 3).

The majority of S2S models show large errors emanating from MJO phase (BMSEp) (Fig. 2b, Supplementary Fig. 4), indicating the inability of current forecasting systems to predict the main location of the MJO realistically[25]. The phase error reduces substantially in all S2S reforecasts by DL-correction (Fig. 2d and Supplementary Fig. 4). The BMSEp averaged over 4 weeks and eight phases is reduced by about 78% after DL-correction in the multi-model mean and by about 45–90% in individual models (Fig. 3). This indicates that, in addition to the amplitude, the MJO location can be better forecasted by applying the DL-correction. Note that when the BMSE is calculated with individual MJO events rather than the composite, the reduction of error is clearly shown as well (Supplementary Fig. 5). Bias correction via the multi-linear regression (MLR, see "Methods" section) model also

reduces the forecast errors (Supplementary Fig. 6), but not as much as the DL-correction. The multi-model mean BMSE from the DL-correction is reduced by about 65% compared to the MLR-correction when averaged over 1 week and eight phases and by 24% over 4 weeks (Supplementary Fig. 6). Particularly, with DL-correction, BMSEa during the first week reduces about 87% compared to the MLR-correction (Supplementary Fig. 6a).

To assess prediction skill and predictability of the MJO, two additional verification metrics are applied (see "Methods" section). The multi-model mean BCOR of the DL-correction is consistently higher than the S2S models up to 4 weeks (Supplementary Fig. 7). An increased signal and reduced noise after the DL-correction, which results in a higher MJO predictability than the original ECMWF-Cy43r3 reforecasts, is also shown (Supplementary Fig. 8).

**Eastward propagation of the predicted MJO.** As mentioned earlier, dynamical forecasts have struggled to accurately forecast the MJO propagation over the Maritime Continent, especially when the forecast is initialized in the Indian Ocean (phases 2 and 3). Such a prediction barrier can be partly explained by the basic state moisture biases that degrade the physical processes associated with the eastward propagation of the MJO[23,24,46,47]. Even the ECMWF-Cy43r3, the best MJO prediction model, simulates an exaggerated Maritime Continent barrier (Supplementary Fig. 2a). Figure 4 shows the reconstructed Outgoing Longwave Radiation (OLR) and zonal wind at 850 hPa (U850) anomalies obtained by projecting the RMMs starting from phase 2 onto the

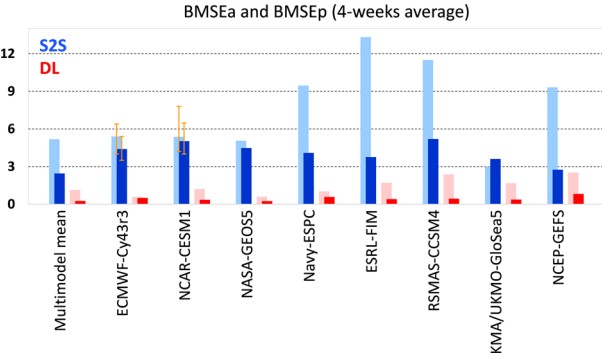

**Fig. 3 Averaged forecast errors in individual models.** Bivariate root-mean-squared amplitude error (BMSEa) (darker colours) and phase error (BMSEp) (lighter colours) for Subseasonal-to-seasonal (S2S) reforecasts (blue) and Deep learning (DL)-corrections (red) averaged over 4 weeks and eight phases shown in Fig. 2, Supplementary Figs. 3 and 4. For ECMWF-Cy43r3 and NCAR-CESM1, the orange error bar denotes the 95% confidence interval based on the bootstrap method. NCEP-GEFS = National Centres for Environmental Prediction Environmental Modelling Centre Global Ensemble Forecast System; NASA-GEOS5 = National Aeronautics and Space Administration Global Modelling and Assimilation Office Goddard Earth Observing System; Navy-ESPC = Naval Research Laboratory Navy Earth System Prediction Capability; RSMAS-CCSM4 = Community Climate System Model version 4 run at the University of Miami Rosenstiel School for Marine and Atmospheric Science; ESRL-FIM = Earth System Research Laboratory Flow-Following Icosahedral Model; NCAR-CESM1 = National Centre for Atmospheric Research Community Earth System Model Version 1; KMA/UKMO-GloSea5 = Korea Meteorological Administration-UK Met Office coupled Global Seasonal forecast; ECMWF-Cy43r3: European Centre for Medium-Range Weather Forecasts version Cy43r3.

normalized eigenvectors used in RMM calculation[45,48]. Compared to the well-organized eastward propagating MJO signal that crosses the Maritime Continent and through the western Pacific in the observations (Fig. 4a), the MJO signal in the ECMWF-Cy43r3 shows fast damping before the convective anomaly reaches the Maritime Continent (~120°E) (Fig. 4b). With DL-correction, however, the MJO anomalies become close to the observations beyond 2 weeks by realistically forecasting both amplitude and phase of the MJO (Fig. 4c). The improved MJO eastward propagation is mostly due to the amplification of the strongly damped MJO signal shown in most of the S2S models (Supplementary Fig. 2).

## Discussion
This study demonstrates the power of Deep learning to be used as a post-processing tool to correct the systematic biases that evolve during MJO forecasts. The errors emanating from MJO amplitude and phase in the dynamical model forecasts are both reduced substantially by DL-correction. The results show that the performance of poor models becomes comparable to the best model after DL-correction. This implies that the differences in the model's performance mainly originate from the systematic errors, rather than the random errors, and the DL method effectively works to minimize them regardless of their amplitude as long as the forecast errors are systematic. Given that huge efforts have been made in operational centres to upgrade their models to reach the level of the world's best model for MJO forecasting (i.e., the ECMWF model), our results show promise for conserving both human and computational resources. Moreover, while the characteristics of systematic biases can change as operational forecast models undergo periodic upgrades, the DL-correction method developed here can be easily adapted to appropriately

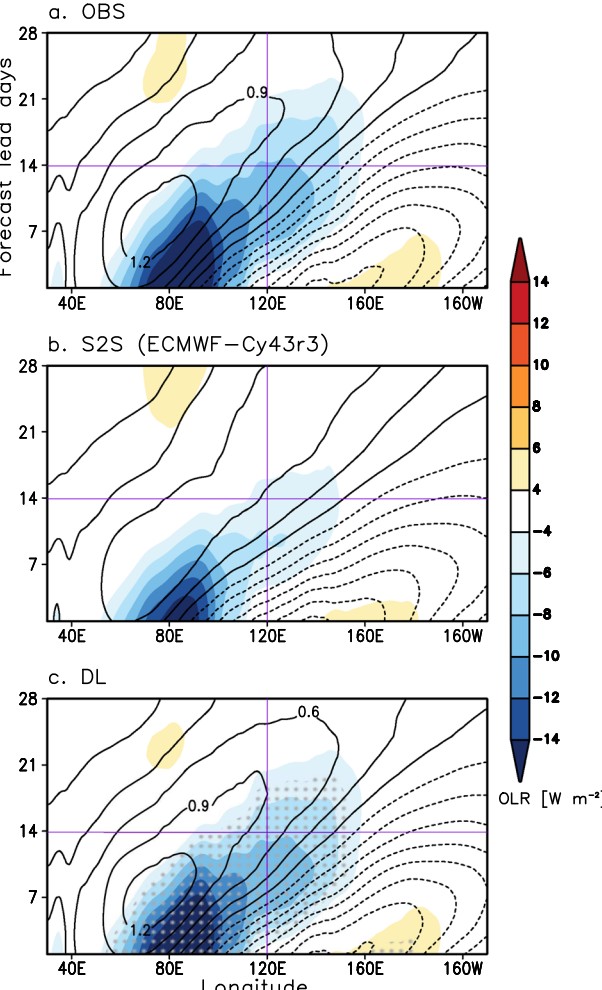

**Fig. 4 Observed and forecasted Madden-Julian oscillation (MJO) propagation.** Reconstructed 15°S–15°N mean Outgoing Longwave Radiation (OLR) (W m⁻², shading) and zonal wind at 850 hPa (U850) (contour interval of 0.3 m s⁻¹) anomalies using the composite of the Real-time Multivariate MJO indices (RMMs) starting from phase 2 shown in Supplementary Fig. 2a for (**a**) observations, (**b**) Subseasonal-to-seasonal (S2S) reforecasts, and (**c**) Deep learning (DL)-corrections. Purple lines indicate day 14 and 120°E. The stipples mark where the OLR anomalies from DL-correction are significantly different from S2S results at the 95% confidence level using the bootstrap method.

reflect the changes. Most importantly, because the model is simple, the approach developed in this study can easily be implemented into real-time MJO forecasts, which in turn can help end-user preparedness and eventually protect lives and properties vulnerable to various hazardous extreme events related to the MJO.

However, although the Deep learning approach can assist in correcting model biases, continuous effort towards developing the dynamical forecast system to minimize the inherent errors is the key for making MJO forecasts reach their potential predictability. Note that the improved MJO prediction with DL-correction does not guarantee an improved prediction of MJO-related phenomena such as tropical cyclones, monsoons, and midlatitude teleconnections, because they rely on both the MJO and the background state within the model. Therefore, further improvements in dynamical models and initialization are fundamental to ultimately improve the S2S predictions.

## Methods

**Hindcasts and validation datasets.** In this study, we use long-term reforecasts from the international Subseasonal-to-Seasonal prediction (S2S[49]) and Subseasonal Experiment (SubX[50]) projects, and from the NCAR Community Earth System Model v1[51], which follows the SubX protocol. Hereafter, we refer to these reforecasts as S2S reforecasts for convenience. Supplementary Table 1 provides information on the eight S2S reforecast models, including initialization interval, ensemble size, reforecast period, and sample size. Note that the reforecast period of ECMWF-Cy43r3 and KMA/UKMO-GloSea5 are different compared to the rest of the models (Supplementary Table 1). These eight models have shown good performance in MJO prediction[24,47].

To identify the MJO events, the Real-time Multivariate MJO (RMM)[45] index is calculated with the daily mean zonal wind at 850 hPa (U850) and 200 hPa (U200) from the ECMWF Interim Reanalysis[52] and Outgoing Longwave Radiation (OLR) from the NOAA Advanced Very High-Resolution Radiometer[53]; these are referred to as observation for brevity. All S2S reforecasts and observations are interpolated onto a 1° longitude and 1° latitude grid. The method for calculating anomalies and RMM indices follows previous studies[24,30].

**Deep learning bias-correction model.** The Deep learning bias correction (hereafter, DL-correction) model utilizes the Long Short-Term Memory (LSTM), which has been proven to be powerful for time sequence modelling[54,55] (Supplementary Fig. 1). It has a cell state ($c_t$), which accumulates the information from the previous states ($t$-1) up to time $t$. The forget gate ($f_t$) controls the extent to which the previous cell state ($c_{t-1}$) is forgotten. The status of an input variable ($X$) at time $t$ is contained in the updated state ($g_t$), and the input gate ($i_t$) determines how much the updated state is retained in the cell state ($c_t$). The cell state ($c_t$) and the updated status ($g_t$) are combined and then propagate into the final state ($h_t$), which is further controlled by the output gate ($o_t$).

In the training period, the input variables ($X$) are the modelled RMM1 and RMM2 indices from the S2S reforecasts, and the output variables ($Y$) are the observed RMM1 and RMM2. We inactivated the cell state $c_{t-1}$ and hidden state $h_{t-1}$ to correct the modelled data at time $t$, to focus on the simultaneous relationship between the input variables (i.e., modelled RMM indices) and the output variables (i.e., observed RMM indices). That is, while the LSTM is often used for predicting the time sequence of the data, we utilized the LSTM to improve the quality of the modelled data by correcting the systematic biases in the S2S models.

The LSTM is trained using the Adaptive Moment Estimation optimizer[56] and mean-square-error loss to optimize weights and biases. Here, the training and validation sets are the same datasets. One to three hidden layers and 3-100 nodes have been tested, while additional hidden layers and nodes did not improve the DL-correction performance. To keep the process as simple and efficient as possible, the final DL-correction model uses one input layer with two nodes, one hidden layer with 10 nodes, and one output layer with two nodes. Note that adding more input variables, such as the leading principal components of OLR and zonal winds, degrades the skill (not shown), hence only RMMs are used for both input and output variables.

**DL-correction procedure.** The leave-one-year-out cross-validation (LOOCV) procedure is often used for making predictions on data not used in the training period and is appropriate for a relatively small dataset. For example, to process DL-correction on the target year 1997 in ECMWF-Cy43r3, the modelled/observed RMMs of MJO events from the rest of 19 years (from 1998 to 2016) are used to train the LSTM model. Then, the weighting coefficients and biases obtained during the training period are directly applied to the modelled RMMs of MJO events in 1997 (target year). This results in DL-corrected MJO predictions in 1997. For the target year 1998, MJO events from the rest of the 19 years (1997 and from 1999 to 2016) are used to train the LSTM model, and so on. The LSTM model is built at every target year, forecast lead time, MJO phase, and each model individually due to their unique systematic biases. Note that, for given input data sets that were randomly selected, the LOOCV produces very similar results for every target year, indicating that the LSTM model is stable.

We also perform the DL-correction in a real forecast manner. The ECMWF-Cy43r3 reforecasts during the first 10 years (1997–2006) are used as the training period to build the LSTM model, and the remaining independent 10 years (2007–2016) are evaluated. In this real forecast procedure, biases are still significantly reduced compared to the raw S2S reforecasts (not shown), but larger than those by the LOOCV approach due to the limited training sample size.

The MJO amplitude (A) and phase (θ) for the observation ($A_o$ and $\theta_o$) and reforecast ($A_m$ and $\theta_m$) are defined as[45,47]:

$$A_o(t) = \sqrt{O_1^2(t) + O_2^2(t)} \tag{1}$$

$$A_m(t, \tau) = \sqrt{M_1^2(t, \tau) + M_2^2(t, \tau)} \tag{2}$$

$$\theta_o(t) = \tan^{-1}\left(\frac{O_2(t)}{O_1(t)}\right) \tag{3}$$

$$\theta_m(t, \tau) = \tan^{-1}\left(\frac{M_2(t, \tau)}{M_1(t, \tau)}\right) \tag{4}$$

where $O_1(t)$ and $O_2(t)$ are the observed RMM1 and RMM2 at time $t$, and $M_1(t, \tau)$ and $M_2(t, \tau)$ are the modelled RMM1 and RMM2 at time $t$ with a lead time of $\tau$ days. On the two-dimensional phase-space diagram[45], the MJO phase is defined as the azimuth of the RMM1 and RMM2 combination and is usually divided into eight phases depending on the location of the MJO convection[45]. The MJO amplitude is determined based on the distance of the azimuth point from the origin, and an MJO event is defined when the observed MJO amplitude ($A_o$) exceeds 1.0 on initial day 0. Although the MJO is most active during boreal winter and thus has the highest forecast skill in this season[25], we use MJO events from all seasons due to the limited sample size. For the same reason, we group two MJO phases (phases 2 & 3, 4 & 5, 6 & 7, and 8 & 1) when training the LSTM model. Note that MJO forecasts of the grouped phases generally possess similar characteristics of errors.

The selected MJO events differ among models due to different initialization frequencies and reforecast periods (Supplementary Table 1). The number of MJO events for initial MJO phases 2 & 3 used for training and target period is listed in Supplementary Table 1 as an example, while other phases show similar event counts. All reforecasts used here are the ensemble mean. Applying the DL-correction to individual ensembles and then averaging the results shows lower performance than applying the DL-correction directly to the ensemble mean (not shown). This indicates that the DL-correction is targeted to reduce the systematic forecast errors, and it is obscured by the random errors in the individual ensemble members.

To establish a baseline for assessing the benefit of the DL-correction method, a multi-linear regression (MLR) model, a standard linear approach for post-processing, is compared. The MLR-correction is identical to the DL-correction in that it corrects RMM1 and RMM2 separately using the modelled RMMs as input and observed RMMs as output with the LOOCV procedure.

**Assessment of MJO predictions.** To evaluate the MJO forecast quality, the bivariate correlation coefficient (BCOR)[25] and bivariate root-mean-squared error (BMSE)[47] are calculated between the predicted and observed RMM indices as a function of forecast lead days as follows:

$$BCOR(\tau) = \frac{\sum_{t=1}^{N}[O_1(t)M_1(t, \tau) + O_2(t)M_2(t, \tau)]}{\sqrt{\sum_{t=1}^{t=N}[O_1^2(t) + O_2^2(t)]}\sqrt{\sum_{t=1}^{t=N}[M_1^2(t, \tau) + M_2^2(t, \tau)]}} \tag{5}$$

$$BMSE(\tau) = \frac{1}{N}\sum_{t=1}^{N}([O_1(t) - M_1(t, \tau)]^2 + [O_2(t) - M_2(t, \tau)]^2) \tag{6}$$

where $N$ is the number of MJO events. The BMSE can be separated into the error emanating from amplitude error (BMSEa) and phase error (BMSEp)[47] as:

$$BMSE(\tau) = BMSEa(\tau) + BMSEp(\tau) \tag{7}$$

$$BMSEa(\tau) = \frac{1}{N}\sum_{t=1}^{N}[A_m(t, \tau) - A_o(t)]^2 \tag{8}$$

$$BMSEp(\tau) = \frac{1}{N}\sum_{t=1}^{N}2A_m(t, \tau)A_o(t)*\{1 - \cos[\theta_m(t, \tau) - \theta_o(t)]\} \tag{9}$$

The MJO potential predictability is assessed via the signal and noise[25] defined as:

$$Signal(\tau) = \frac{1}{N}\sum_{t=1}^{N}(\overline{M_1(t, \tau)}^2 + \overline{M_2(t, \tau)}^2) \tag{10}$$

$$Noise(\tau) = \frac{1}{N}\sum_{t=1}^{N}(\overline{M_1'(t, \tau)}^2 + \overline{M_2'(t, \tau)}^2) \tag{11}$$

where overbar denotes ensemble mean and prime presents individual ensembles' deviations from the ensemble mean. In this formulation, the signal refers to the variability of the ensemble mean while the noise refers to the variability of individual forecasts around the ensemble mean (i.e., the forecast spread), and both quantities depend on the forecast lead time ($\tau$). The ECMWF-Cy43r3 is used to estimate the signal and noise due to its relatively large ensemble size and high MJO skill.

**Confidence interval.** The statistical significance test is performed with ECMWF-Cy43r3 and NCAR-CESM1 only, due to their relatively large ensemble sizes. The confidence interval of DL-correction results is calculated using the bootstrap method. We randomly select 11 ensemble members from the S2S reforecasts with allowing overlap to calculate the ensemble-averaged BMSE. This process is repeated 10,000 times and the 2.5th and 97.5th percentile values are used to define the 95% confidence interval. The ensemble-averaged BMSE of DL-correction value (Fig. 3, Supplementary Figs. 3, 4) is significant at the 95% confidence level if it lies outside the 2.5th or 97.5th percentile. The same process is performed for the

reconstructed OLR and U850 to check whether the composited anomalies from DL-correction is significantly different from the S2S forecast results (Fig. 4).

## Data availability

Data related to this paper can be downloaded from: ERA-Interim, http://apps.ecmwf.int/datasets/data/interim_full_daily; NOAA OLR, https://www.esrl.noaa.gov/psd/data/gridded/data.interp_OLR.html; The SubX and NCAR-CESM1 reforecasts, http://iridl.ldeo.columbia.edu/SOURCES/.Models/.SubX; The S2S reforecasts, https://apps.ecmwf.int/datasets/data/s2s/. The data that support the findings of this study are available at https://zenodo.org/record/4601794 and from the corresponding author upon reasonable request.

## Code availability

TensorFlow (https://www.tensorflow.org) libraries were implemented to formulate the forecast model using the LSTM. The codes used in the current analysis are available at https://zenodo.org/record/4601794 and from the corresponding author upon reasonable request.

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

## Acknowledgements

H.K. was supported by NSF grant AGS-1652289 and the Brain Pool program funded by the Ministry of Science and ICT through the National Research Foundation of Korea (NRF) 2019H1D3A2A01102234. Y.G.H. and Y.S.J. are supported by the NRF under Grant No. NRF-2020R1A2C2101025. S.W.S. was supported by the Korea Meteorological Administration Research and Development Program under Grant KMI2020-01010.

## Author contributions

H.K. designed the research, performed model run, analyzed the model output, and generated figures. H.K., Y.G.H. and Y.S.J. built the LSTM model. H.K., Y.G.H., Y.S.J. and S.W.S. discussed the results and contributed to the writing of the manuscript.

## Competing interests

The authors declare no competing interests.
