## [Peer Review File · Nature Communications]

Reviewers' Comments:

Reviewer #1:

Remarks to the Author:

Please see my comments in the attached document. I suggest minor revisions for the publication of this high impact work in the journal. I enjoyed reading the work very much and learning about the methodology used.

Deep Learning for MJO Prediction

Kim et al., 2020

The authors blend dynamical model forecasts with a Deep Learning model to improve prediction skill of the Madden-Julian Oscillation (MJO). The MJO serves as a primary source of subseasonal predictability for the global climate system. With Deep Learning post-processing, the authors show that the multi-model forecast errors in MJO amplitude and phase averaged over four weeks are reduced by about 90% and 77%, respectively. Their deep learning model even improves the MJO propagation through the maritime continent, which has been a long-standing challenge for weather and climate models to get right.

The paper is well organized and clear. The results are definitely significant and a step change in weather prediction field by combining machine learning with weather prediction models. Once the minor revisions below are addressed, I feel the manuscript would be a welcome addition to the MJO prediction community. The manuscript will be suitable for publication in the Nature Communications journal after the suggested revisions.

Comments:

General comment: There are some typos that need to be corrected before the manuscript can be ready for publication. There is good hindcast verification in the manuscript and the authors quantify improved skill in a valuable manner, yet the conclusions and discussion are a bit vague and not based on quantified metrics about why two variables would co-evolve and how they can be co-predicted with improved skill. Improving the discussion of the paper and eliminating the simple typos will make it a much stronger paper.

Comments:

Following line can be split into two for easier reading: “Model deficiencies in simulating realistic MJO events are partially due to our poor understanding of the underlying physics, thus more efforts on process-level diagnostics are suggested to further improve MJO simulation and prediction”

Change preposition “at” to “on” in this sentence: “A large discrepancy between S2S reforecasts and observations is clearly shown on day 1.”

Could the authors present a hypothesis of why the MJO propagation across the Maritime Continent is improved with the DL model? They only mention “With DL-correction, the MJO anomalies become close to the observations beyond two weeks by realistically forecasting both amplitude and phase” with no explanation or possible hypothesis. It would be good to present a hypothesis here that could be used for model development if possible.

Fig. 2: Why show an anomalous color bar with positive and negative colors if all

Deep Learning for MJO Prediction

Kim et al., 2020

values are positive lead time? The authors could just show a positive lead days colorbar and mention in the caption that there are no negative days.

For the LSTM training, will the LOOCV methodology bias the LSTM models to foresee longterm variability in the RMM modes (multiyear/decadal variability) which could help the predictions for the year left out?

Reviewer #2:

Remarks to the Author:

Comments on 'Deep learning for MJO prediction' by Kim et al. submitted to Nature Communications.

Skillful subseasonal prediction has a direct benefit to the society and economy, while it remains very challenging. The Madden-Julian Oscillation (MJO) acts as one of the major predictability sources and received great attention from the community. The MJO prediction skill in the state-of-the-art dynamic models differs dramatically, with some models have skill about two weeks and the ECMWF model has an outstanding skill beyond 30 days. To improve the MJO prediction skill is one of the major tasks for the S2S prediction community. The Deep Learning (DL) post-processing leads to an improved amplitude and phase prediction of MJO compared to the raw model results. I think this is an interesting and timely study, and provide a new path for S2S prediction. I would recommend this paper to be considered for publication if the authors can address my following comment as listed below.

1) As a predictor, the MJO is emphasized because it provides predictability sources for subseasonal predictions. The DL technique may lead to an 'improved' MJO prediction through the post-processing, while it may not benefit the real prediction of other variables and phenomena, such as temperature, precipitation, TC, atmospheric rivers, and so on. The modulation of MJO on other phenomena relies on the 'actual' amplitude/phase of MJO and the background mean climate within the dynamic models. The 'improved' MJO prediction, in particular through the post-process, does not guarantee an improved prediction of its impacts. It also implies that the further improvement of dynamic models and initialization tends to be fundamental to improve the S2S predictions. From this point of view, this work is one example to show the potential application of the proposed methodology on subseasonal prediction. However, I think this work itself is not significant enough to warrant its publication in Nature Communications unless this can be demonstrated to some extent. One way is to build a hybrid dynamic-statistical model (using the MJO in model prediction and the observational relationship between MJO and other climate variables, such as temperature, precipitation, TC, circulation) to truly prove the value of the DL on S2S prediction (one example is fine).

2) I am surprised that 'day-1' MJO shows a relatively large bias from Fig. 1. Compared to the first author's previous work using the ECMWF and NCEP models (Fig. 9, Kim et al. 2014), the bias shown here seems to be too big. How to understand this? Please clarify.

3) Here the authors only show the results of RMSE. I suggest showing the anomalous correlation coefficient as well.

4) I suggest add some discussion about the implication of this study on model development to improve the MJO simulation and prediction. Also, it is better to provide some insights about why the individual models using DL show similar skills, no matter how bad or good the raw model is?

5) Does the DL post-processing change the potential predictability? It will be interesting to check this.

6) 'The LSTM model is built at every target year, forecast lead time, MJO phase, and each model individually due to their unique systematic biases'. Does the LSTM differ much for different years? The leave-one-year-out cross-validation is shown, while it will be useful to show the 'real' prediction. For example, the data during the first 10 years are used as the training period to build the LSTM model, the left independent 7/10 years are examined to show its application on MJO prediction.

Reviewer #3:

Remarks to the Author:

The work presents a method of bias correction for MJO forecasts using deep learning. The main idea is to train an algorithm that maps modelled RMM indices to observed values. Then apply the mapping to RMM forecasts to obtain corresponding bias corrected forecasts. The method is well describes and appears to work well. Therefore the work is worth publishing in my judgement. However, I have some reservations as described below

1. The work is not on the use of deep learning to make MJO forecasts rather to correct biases on forecasts made by dynamical models. That should be clear in the title and abstract. For example the

title could be " Application of deep learning in the bias correction of MJO forecasts"

2. There is little or no mention of other methods of bias correction. Modeling centers are typically aware of biases and drifts in their models and have methods for correcting/accounting for them. This work needs to be put in that context. Specifically how does it compare with other methods?

3. The key challenge the that authors did not address is uncertainty. How is the error-bar/confidence level affected by the bias correction?

Manuscript title: “Deep Learning for bias correction of MJO prediction”

Authors: H. Kim, Y. G. Ham, Y. S. Joo, S. W. Son

* We wish to express our gratitude to the reviewers for the time and effort spent in reviewing our manuscript and we feel the updated version is much stronger. The major changes include:

- Adding confidence interval to Figure 3, Figure 4, Extended Data Fig. 3 and Fig. 4.
- Prediction, predictability metrics: new Extended Data Fig. 7 (BCOR) and Fig. 8 (Signal and noise)

Reviewer #1

The authors blend dynamical model forecasts with a Deep Learning model to improve prediction skill of the Madden-Julian Oscillation (MJO). The MJO serves as a primary source of subseasonal predictability for the global climate system. With Deep Learning post-processing, the authors show that the multi-model forecast errors in MJO amplitude and phase averaged over four weeks are reduced by about 90% and 77%, respectively. Their deep learning model even improves the MJO propagation through the maritime continent, which has been a long-standing challenge for weather and climate models to get right.

The paper is well organized and clear. The results are definitely significant and a step change in weather prediction field by combining machine learning with weather prediction models. Once the minor revisions below are addressed, I feel the manuscript would be a welcome addition to the MJO prediction community. The manuscript will be suitable for publication in the Nature Communications journal after the suggested revisions.

General comment:

- There are some typos that need to be corrected before the manuscript can be ready for publication. There is good hindcast verification in the manuscript and the authors quantify improved skill in a valuable manner, yet the conclusions and discussion are a bit vague and not based on quantified metrics about why two variables would coevolve and how they can be co-predicted with improved skill. Improving the discussion of the paper and eliminating the simple typos will make it a much stronger paper.

→ Thanks for your constructive comments. We did our best to improve the discussion and correct typos throughout the entire manuscript.

Comments:

(1) Following line can be split into two for easier reading: “Model deficiencies in simulating realistic MJO events are partially due to our poor understanding of the underlying physics, thus more efforts on process-level diagnostics are suggested to further improve MJO simulation and prediction”

→ Split into two sentences as suggested.

(2) Change preposition “at” to “on” in this sentence: “A large discrepancy between S2S reforecasts and observations is clearly shown on day 1.”

→ Changed as suggested throughout the entire manuscript.

(3) Could the authors present a hypothesis of why the MJO propagation across the Maritime Continent is improved with the DL model? They only mention “With DL correction, the MJO anomalies become close to the observations beyond two weeks by realistically forecasting both amplitude and phase” with no explanation or possible hypothesis. It would be good to present a hypothesis here that could be used for model development if possible.

→ We think the main reason for the improvement is because the DL-correction amplifies the strongly damped MJO signal in the original forecasts (Fig. 4). The S2S/SubX models tends to underestimate the amplitude of the MJO especially over the maritime continent (Kim et al. 2019), and this systematic error is likely to be successfully captured, therefore, corrected by the DL model. Since this is a statistical correction of RMMs, it is hard to link the improved propagation with any physics-based hypothesis, such as moisture advection processes, etc. We make our point clear and add the implication on model development as:

(Page 4) “With DL-correction, however, the MJO anomalies become close to the observations beyond two weeks by realistically forecasting both amplitude and phase of the MJO (Fig. 4c). The improved MJO eastward propagation is mostly due to the amplification of the strongly damped MJO signal shown in most of the S2S models (Extended Data Fig. 2).”

(Page 4) “However, although Deep Learning approach can assist in correcting model biases, continuous effort on developing the dynamical forecast system to minimize the inherent errors is the key for making MJO forecast to reach its potential predictability. Note that the improved MJO prediction with DL-correction does not guarantee an improved prediction of MJO-related phenomena such as tropical cyclones, monsoons, and midlatitude teleconnections, because they rely on both MJO and background state within the model. Therefore, further improvement of dynamical model and initialization are fundamental to ultimately improve the S2S predictions.”

(reference) H. Kim, M. A. Janiga, K. Pegion, MJO Propagation Processes and Mean Biases in the SubX and S2S Reforecasts. *Journal of Geophysical Research: Atmospheres* 124, 9314-9331 (2019)

(4) Fig. 2: Why show an anomalous color bar with positive and negative colors if all values are positive lead time? The authors could just show a positive lead days color bar and mention in the caption that there are no negative days.

→ Changed to positive-only color bar as suggested. Thanks for pointing this out.

(5) For the LSTM training, will the LOOCV methodology bias the LSTM models to foresee longterm variability in the RMM modes (multiyear/decadal variability) which could help the predictions for the year left out?

→ To examine whether the RMM indices have multiyear/decadal variability, Figure A shows the power spectrum of the observed daily RMM1 index from 1979 to 2016. As expected, the RMM1 has significant power on the intraseasonal period (30~80 days), but not on periods longer than the intraseasonal timescale window. Therefore, we can conclude that the forecasts via LSTM models are not influenced by long-term MJO variability.

Figure A: Power spectra of the daily RMM1 index over the period of 19790101 to 20161231. Shown are also the Markov red noise spectrum (black curve) and its bounds at confidence level of 5% and 95% (black dashed curves). The two vertical red lines represent the 30~80 days range.

As an alternative way to examine the year-to-year variation and stability of the LSTM model, we compare the prediction results of each target year. Specifically, we target forecasting MJO events (phase 2 and 3) selected from random years from 1997 to 2016 with allowing duplication of years. A total of 25 MJO events are randomly selected from 20 years. Then, we use the LSTM model built in our study for each year (LSTM(yr)) to predict the 25 events. If forecast results from each LSTM(yr) show consistent forecasts for the randomly selected 25 MJO events, it indicates that the model does not differ much for different years, and model is stable.

- Figure B shows the RMM1 index on forecast day-21 (note that this is consistent in other lead days). Although some variation exists, the year-to-year variation of RMM1 forecast in each 25 MJO cases are consistent for 20 years.

Response to the reviewers' comments

- Figure C shows the 4-week forecast of the RMM1 index from randomly selected case among 25 cases (MJO case from year 2016). The 20 years show consistent results of forecasting this specific case.

A discussion is now added in the method section as:

(Page 13): “Note that, for a given input data sets that were randomly selected, the LOOCV produce very similar results for every target year, indicating that the LSTM model is stable.”

Figure B: RMM1 forecast for forecast lead day-21 for 25 randomly selected MJO phase2&3 cases as a function of target years.

Figure C: RMM1 index for a specific case (2016 case) in observation (black), raw ECMWF (blue), and DL-correction (red). The red contour line represents the average of 20 years (1997-2016), the gray box outlines the ± 1.0 standard deviation of RMM1, and whiskers

Response to the reviewers' comments

indicate the minimum and maximum RMM1 values among 20 different LSTM model results.

Reviewer #2

Skillful subseasonal prediction has a direct benefit to the society and economy, while it remains very challenging. The Madden-Julian Oscillation (MJO) acts as one of the major predictability sources and received great attention from the community. The MJO prediction skill in the state-of-the-art dynamic models differs dramatically, with some models have skill about two weeks and the ECMWF model has an outstanding skill beyond 30 days. To improve the MJO prediction skill is one of the major tasks for the S2S prediction community. The Deep Learning (DL) post-processing leads to an improved amplitude and phase prediction of MJO compared to the raw model results. I think this is an interesting and timely study and provide a new path for S2S prediction. I would recommend this paper to be considered for publication if the authors can address my following comment as listed below.

→ Thanks for your constructive comments. We tried our best to address your comments.

(1) As a predictor, the MJO is emphasized because it provides predictability sources for subseasonal predictions. The DL technique may lead to an 'improved' MJO prediction through the post-processing, while it may not benefit the real prediction of other variables and phenomena, such as temperature, precipitation, TC, atmospheric rivers, and so on. The modulation of MJO on other phenomena relies on the 'actual' amplitude/phase of MJO and the background mean climate within the dynamic models. The 'improved' MJO prediction, in particular through the post-process, does not guarantee an improved prediction of its impacts. It also implies that the further improvement of dynamic models and initialization tends to be fundamental to improve the S2S predictions.

From this point of view, this work is one example to show the potential application of the proposed methodology on subseasonal prediction. However, I think this work itself is not significant enough to warrant its publication in Nature Communications unless this can be demonstrated to some extent. One way is to build a hybrid dynamic-statistical model (using the MJO in model prediction and the observational relationship between MJO and other climate variables, such as temperature, precipitation, TC, circulation) to truly prove the value of the DL on S2S prediction (one example is fine).

→ We appreciate the reviewer's valuable points. First of all, we totally agree that the improved MJO prediction does not guarantee an improved prediction of its impacts, so we reflect this point in the revision as:

(Page 4) "Note that the improved MJO prediction with DL-correction does not guarantee an improved prediction of MJO-related phenomena such as tropical cyclones, monsoons, and midlatitude teleconnections, because they rely on both MJO and background state within the model. Therefore, further improvement of dynamical model and initialization are fundamental to ultimately improve the S2S predictions."

Forecasting the MJO itself is far from perfect, and the best model being still 2-3 weeks lower than the potential predictability. So, we would like to firstly demonstrate the usefulness of deep learning for MJO only, since MJO prediction alone presents considerable systematic biases.

While we totally agree with the reviewer, ultimately though, we felt a look at MJO impact is outside the scope of this study and is more suited for a stand-alone paper. As suggested, our next plan is to apply the DL-correction to the prediction of MJO-related Atmospheric Rivers (Zhou et al. 2021), PNA patterns (Wang et al. 2020a, b), East Asia precipitation (Kim et al. 2020), and Air Pollution (Jung et al., under review) in which we found a strong relationship with the MJO.

- Zhou, Y., Kim, H., & Waliser, D. E. (2021). Atmospheric river lifecycle responses to the Madden-Julian Oscillation. *Geophysical Research Letters*, 48, <https://doi.org/10.1029/2020GL090983>
- Wang, J., H. Kim, D. Kim, S. A. Henderson, C. Stan, E. D. Maloney, 2020a: MJO teleconnections over the PNA region in climate models. Part II: Impacts of the MJO and basic state, *J. Climate*, 33, 5081–5101, <https://doi.org/10.1175/JCLI-D-19-0865.1>
- Wang, J., H. Kim, D. Kim, S. A. Henderson, C. Stan, E. D. Maloney, 2020b: MJO teleconnections over the PNA region in climate models. Part I: Performance- and process-based skill metrics, *J. Climate*, 33, 1051–1067 [10.1175/JCLI-D-19-0253.1](https://doi.org/10.1175/JCLI-D-19-0253.1)
- Kim, H., S.-W. Son, and C. Yoo, 2020: QBO modulation of the MJO-related precipitation in East Asia, *Journal of Geophysical Research - Atmosphere*, 125, <https://doi.org/10.1029/2019JD031929>.
- Jung, M. I., S. W. Son, H. Kim, D. Chen: Tropical modulation of East Asia air pollution (under review).

(2) I am surprised that ‘day-1’ MJO shows a relatively large bias from Fig. 1. Compared to the first author’s previous work using the ECMWF and NCEP models (Fig. 9, Kim et al. 2014), the bias shown here seems to be too big. How to understand this? Please clarify.

→ Figure D compares the results: original figure from Kim et al. (2014) (left panel) and recent version in this paper (right panel) with some modifications for fair comparison (the format and color (in red) as well as the MJO selection criteria (RMM amplitude >1.5)). The weaker amplitude of the RMM indices is evident in both forecasts, which denotes that the overall bias shown in the current study is not systematically larger than that in Kim et al. (2014). The difference could be due to different reforecast periods (1993-2009 vs. 1997-2016), model versions (Cy32 vs. Cy43), etc., while it is not easy to disentangle the exact reason.

Figure D: RMM composites for initial phases 1-3-5-7 from (left) Kim et al. 2014 and (right) current study.

(3) Here the authors only show the results of RMSE. I suggest showing the anomalous correlation coefficient as well.

→ Thanks for your suggestion. We add the multi-model averaged bivariate correlation coefficient (BCOR) of MJO predictions before and after DL-correction (new Extended Data Fig. 7 and methodology) and add a discussion as:

(Page 3) “To assess prediction skill and predictability of the MJO, two additional verification metrics are applied. It turns out that the multi-model mean BCOR of the DL-correction is consistently higher than the S2S models up to four weeks (Extended Data Fig. 7). The increased signal and reduced noise after the DL-correction, resulting in a higher MJO predictability than the original ECMWF-Cy43r3 reforecasts, is also observed (Extended Data Figure 8).”

Extended Data Fig. 7: Correlation coefficients in multi-model mean S2S reforecasts and DL-corrections. Multi-model mean BCOR of S2S reforecasts (blue) and DL-corrections (red). The gray horizontal line is BCOR of 0.6.

(4) I suggest add some discussion about the implication of this study on model development to improve the MJO simulation and prediction. Also, it is better to provide some insights about why the individual models using DL show similar skills, no matter how bad or good the raw model is?

→ As suggested, we add a discussion as:

(Page 4) “The highlight of the results is that the performance of poor models becomes comparable to the best model after DL-correction. This implies that the differences in the model’s performance are mainly originated from the systematic errors, rather the random errors, and the DL method effectively works to minimize them with regardless of their amplitude as far as the forecast errors are systematic.”

(Page 4) “However, although Deep Learning approach can assist in correcting model biases, continuous effort on developing the dynamical forecast system to minimize the inherent errors is the key for making MJO forecast to reach its potential predictability. Note that the improved

MJO prediction with DL-correction does not guarantee an improved prediction of MJO-related phenomena such as tropical cyclones, monsoons, and midlatitude teleconnections, because they rely on both MJO and background state within the model. Therefore, further improvement of dynamical model and initialization are fundamental to ultimately improve the S2S predictions.”

To address the following comments #5 and #6, we use the ECMWF-Cy43r3 reforecast because it has (i) the best MJO forecast skill, (ii) the longest period of hindcast data (1997-2016, 20 years), (iii) frequent initialization (twice per week), and (iv) large number of ensemble members (11 ensemble members). Note that we stored the reforecast output of individual ensemble members from ECMWF and CESM1 only due to the lack of computing resources.

(5) Does the DL post-processing change the potential predictability? It will be interesting to check this.

→ Thank you for the suggestion. Figure below shows the signal and noise from the raw ECMWF hindcast and DL-correction (new Extended Data Fig. 8). Clearly, the signal is larger, and noise is smaller after DL-correction, making the potential predictability higher. The methodology and discussion are now added as:

(method): “The MJO potential predictability is assessed via the signal and noise²⁵ defined as:

$$\text{Signal } (\tau) = \frac{1}{N} \sum_{t=1}^N (\overline{M_1(t, \tau)}^2 + \overline{M_2(t, \tau)}^2)$$

$$\text{Noise } (\tau) = \frac{1}{N} \sum_{t=1}^N (\overline{M'_1(t, \tau)}^2 + \overline{M'_2(t, \tau)}^2)$$

where overbar denotes ensemble mean and prime presents individual ensembles' deviations from the ensemble mean. In this formulation, the signal refers to the variability of ensemble mean while the noise refers to the variability of individual forecasts around the ensemble mean (i.e., the forecast spread), and both quantities depend on the forecast lead time (τ). The ECMWF-Cy43r3 is used to estimate the signal and noise due to its relatively large ensemble size and high MJO skill.”

(Page 3): “To assess prediction skill and predictability of the MJO, two additional verification metrics are applied. It turns out that the multi-model mean BCOR of the DL-correction is consistently higher than the S2S models up to four weeks (Extended Data Fig. 7). The increased signal and reduced noise after the DL-correction, resulting in a higher MJO predictability than the original ECMWF-Cy43r3 reforecasts, is also observed (Extended Data Figure 8).”

Extended Data Fig. 8: Forecast signal and noise. Signal (solid) and noise (dashed) as a function of forecast lead days for ECMWF-Cy43r3 (blue) and DL-correction (red).

(6) ‘The LSTM model is built at every target year, forecast lead time, MJO phase, and each model individually due to their unique systematic biases’. Does the LSTM differ much for different years?

→ To examine the stability of the LSTM model, we compare the prediction results of each target year. In this case, the input variables are set to be equal in all years. Firstly, we target forecasting MJO events (phase 2 and 3) selected from random years from 1997 to 2016 with allowing duplication of years. A total of 25 MJO events are randomly selected from 20 years. Then, we use the LSTM model built in our study for each year (LSTM(yr)) to predict the 25 events. If forecast results from each LSTM(yr) show consistent forecasts for the randomly selected 25 MJO events, it indicates that the model does not differ much for different years, and model is stable.

- Figure E shows the RMM1 index on forecast day-21 (note that this is consistent in other lead days). Although some variation exists, the year-to-year variation of RMM1 forecast in each 25 MJO cases are consistent for 20 years.
- Figure F shows the 4-week forecast of the RMM1 index from randomly selected case among 25 cases (MJO case from year 2016). The 20 years show consistent results of forecasting this specific case.

A discussion is now added in the method section as:

(Page 13): “Note that, for a given input data sets that were randomly selected, the LOOCV produce very similar results for every target year, indicating that the LSTM model is stable.”

Figure E: RMM1 forecast for forecast lead day-21 for 25 randomly selected MJO phase2&3 cases as a function of target years.

Figure F: RMM1 index for a specific case (2016 case) in observation (black), raw ECMWF (blue), and DL-correction (red). The red contour line represents the average of 20 years (1997-2016), the gray box outlines the ± 1.0 standard deviation of RMM1, and whiskers indicate the minimum and maximum RMM1 values among 20 different LSTM model result.

(7) The leave-one-year-out cross-validation is shown, while it will be useful to show the ‘real’ prediction. For example, the data during the first 10 years are used as the training period to build the LSTM model, the left independent 7/10 years are examined to show its application on MJO prediction.

→ Thanks for the suggestion. To see whether the DL-correction is useful for the real prediction, we perform two experiments with different training/testing sets from ECMWF-Cy43r3:

(A) **10y-train forecast**: data during the first 10 years (1997-2006) are used as the training period to build the LSTM, and the left independent 10 years (2007-2016) are evaluated.

(B) **Real forecast:** 2007-2016 are evaluated. However, the training period extends to the year prior to the target year. For example, for predicting 2007 MJO, 1997-2006 is used for training. For 2008 MJO forecast, 1997-2007 is used, and so on.

Since both methods (A) and (B) are evaluated against 2007-2016, we compare those results with our original method (LOOCV) over the same period, 2007-2016. The BMSEa is compared in Figure G. It is shown that any of these methods result in reduced forecast errors, compared to raw ECMWF forecasts, while the level of reduction differs by the training method mainly due to the number of samples. Note that the results from BMSEp are similar as well (not shown). To reflect the reviewer's comments on the revision, we add discussion as:

(Method, page 13) "We also perform the DL-correction in a real forecast manner. The ECMWF-Cy43r3 reforecasts during the first 10 years (1997-2006) are used as the training period to build the LSTM model, and the left independent 10 years (2007-2016) are evaluated. In this real forecast procedure, biases are still significantly reduced compared to the raw S2S reforecasts (not shown), but larger than those by the LOOCV approach due to the limited training sample size."

Figure G: BMSEa for (left) ECMWF reforecasts, (middle) DL-corrections, and (right) their differences with using different training periods.

Reviewer #3

The work presents a method of bias correction for MJO forecasts using deep learning. The main idea is to train an algorithm that maps modelled RMM indices to observed values. Then apply the mapping to RMM forecasts to obtain corresponding bias corrected forecasts. The method is well described and appears to work well. Therefore, the work is worth publishing in my judgement. However, I have some reservations as described below.

→ Thanks for your constructive comments. We tried our best to address your comments.

(1) The work is not on the use of deep learning to make MJO forecasts rather to correct biases on forecasts made by dynamical models. That should be clear in the title and abstract. For example the title could be "Application of deep learning in the bias correction of MJO forecasts"

→ Thanks for your suggestion. The title is now changed to "Deep Learning for bias correction of MJO prediction" and we have made this point clear in the manuscript as suggested.

(2) There is little or no mention of other methods of bias correction. Modeling centers are typically aware of biases and drifts in their models and have methods for correcting/accounting for them. This work needs to be put in that context. Specifically, how does it compare with other methods?

→ Thanks for your suggestion. To address your question, we compare our DL-correction results with a multi-linear regression model. Methodology and discussion are included as:

(method) "To establish a baseline for assessing the benefit of the DL-correction method, a multi-linear regression (MLR) model, a standard linear approach for post-processing, is compared. The MLR-correction is identical to the DL-correction in that it corrects RMM1 and RMM2 separately using the modelled RMMs as input and observed RMMs as output with the LOOCV procedure."

(Page 3) "Bias correction via the multi-linear regression (MLR) model also reduces the forecast errors (Extended Data Fig. 6), but not as much as the DL-correction. The multi-model mean BMSE from the DL-correction is reduced by about 65% compared to the MLR-correction when averaged over one week and eight phases and by 24% over four weeks (Extended Data Fig. 6). Particularly, with DL-correction, BMSE_a during the first week reduces about 87% compared to the MLR-correction (Extended Data Fig. 6a)."

Extended Data Fig. 6: Forecast errors in DL-corrections and MLR-corrections. Same as Fig. 2, except for (a) BMSEa and (b) BMSEp averaged over eight MJO phases in DL-corrections (red) and MLR-corrections (gray).

Also, we contacted researchers from two operational centers which provide MJO real-time forecast to check whether they apply any bias correction: ECMWF and NOAA CPC. ECMWF provides a mean-bias corrected forecast by removing the model climatology (via personal communication with Frederic Vitart, ECMWF). In NOAA CPC forecast, NCEP EMC GEFS provides a version of Bias-Corrected Ensemble Forecast (named as “NCEP EMC GEFSBC”). EMC's bias correction is using a hybrid method which means a combined bias from the latest prior forecast bias for short lead-time, and model climatology for extended range and longer lead-time (via personal communication with Yuejian Zhu, Qin Ginger Zhang, Kyle MacRitchie, and Dan Collins, NOAA). Therefore, as far as we know, there is no bias correction applied to operational MJO forecasts besides removing the model climatological bias which is already performed in our current S2S reforecasts.

* MJO operational forecast page:

ECMWF: <https://www.ecmwf.int/en/forecasts/charts/catalogue/>

NOAA CPC: <https://www.cpc.ncep.noaa.gov/products/precip/CWlink/MJO>

(3) The key challenge the that authors did not address is uncertainty. How is the error-bar/confidence level affected by the bias correction?

→ Thanks for your suggestion. To address this, we utilize 11 ensemble members from each of ECMWF and CESM1 and use bootstrapping. Confidence interval is now added to Figure 3, Figure 4, Extended Data Fig. 3 and Fig. 4. The methodology is added in the text and in figure captions:

(method, page 15) “The statistical significance test is performed with ECMWF-Cy43r3 and NCAR-CESM1 only due to their relatively large ensemble size. The confidence interval of DL-correction results is calculated using the bootstrap method. We randomly select 11 ensembles

Response to the reviewers' comments

members from the S2S reforecasts with allowing overlap to calculate the ensemble-averaged BMSE. This process is repeated 10,000 times and the 2.5th and 97.5th percentile values are used to define the 95% confidence interval. The ensemble-averaged BMSE of DL-correction value (Fig. 3, Extended Data Fig. 3, 4) is significant at 95% confidence level if it lies outside the 2.5th and 97.5th percentile. Same process is performed for the reconstructed OLR and U850 to check whether the composited anomalies from DL-correction is significantly different from the S2S forecast results (Fig. 4).”

Reviewers' Comments:

Reviewer #1:

Remarks to the Author:

The authors have responded to my comments satisfactorily and have substantially improved the manuscript. Thanks for this exciting contribution. I enjoyed learning about this work by reading this well-written manuscript. -Aneesh

Reviewer #2:

Remarks to the Author:

First, I should applaud the authors' effort to address my comments. The revised manuscript is much more clear and improved. However, I still have one comment following my previous comment #3:

Since the error of amplitude and phase has reduced by more than 70%, why the BCOR skill does not change much as shown in the extended Fig. 7? More specifically, the useful prediction skill can be estimated when BCOR drops to 0.5, which is generally corresponding to that when the RMSE increases to $\sqrt{2}$. It is not the case by comparing extended Fig. 5 and 7. This needs some clarification.

Reviewer #3:

Remarks to the Author:

The authors have addressed my questions and concerns. The paper is acceptable for publication in my judgement.

Manuscript title: "Deep Learning for bias correction of MJO prediction"

Authors: H. Kim, Y. G. Ham, Y. S. Joo, S. W. Son

Reviewer #2

- First, I should applaud the authors' effort to address my comments. The revised manuscript is much more clear and improved.

→ We appreciate the reviewer for another careful review.

- However, I still have one comment following my previous comment #3: Since the error of amplitude and phase has reduced by more than 70%, why the BCOR skill does not change much as shown in the extended Fig. 7?

→ First of all, the ~70% reduced errors shown in Fig. 2 and Fig. 3 is calculated based on the 'composite' maps shown in Figure 1. This has been described in the manuscript as (page 3) "To evaluate the forecast errors quantitatively, the bivariate root-mean-squared error (BMSE) is calculated as a function of initial MJO phases **from the composites shown in Fig. 1.**" and also in Figure captions (Fig. 2) "BMSE is divided into (a, c) BMSEa and (b, d) BMSEp for RMMs **composite shown in Fig. 1**". Therefore, the reduced mean errors do not exactly match with the BCOR (Supplementary Fig. 7) which is calculated with all **individual MJO events**.

- More specifically, the useful prediction skill can be estimated when BCOR drops to 0.5, which is generally corresponding to that when the RMSE increases to $\sqrt{2}$. It is not the case by comparing extended Fig. 5 and 7. This needs some clarification.

→ For fair comparison between BMSE (same as RMSE) and BCOR, one should compare the Supplementary Figure 5 and 7 in which the **individual MJO events** are used for BMSE and BCOR calculations. For easier comparison, we combine the Extended Fig. 5 and 7 to compare the multi-model averaged BMSE and BCOR (Figure A below). Clearly, the BCOR and BMSE shows improved skill after DL-correction.

Although previous studies used the BCOR=0.5 and BMSE=1.414 as a threshold for useful MJO skill, the BCOR=0.5 does not always match with the BMSE=1.414 line. For example, Figure B is from Lim et al. (2018) where they compared the BCOR and BMSE of RMM indices using 10 S2S models. The BMSE=1.414 lines do not exactly match with the BCOR=0.5 line as well, so they used the BMSE=2 as a reference skill measure.

Lim, Y., Son, S., & Kim, D. (2018). MJO Prediction Skill of the Subseasonal-to-Seasonal Prediction Models, *Journal of Climate*, 31(10), 4075-4094.

As mentioned by the reviewer, the BCOR does not show continuous improvement after 3 weeks while the BMSE keeps reducing. This could be because the absolute error of RMMs is reduced, but the sign of the error is not as much improved as the absolute error when individual MJO events are considered. This could depend on each forecast systems, and a more detailed analysis on individual model output should be performed to get a clearer picture.

Figure A: Multi-model averaged BMSE from individual MJO events (dashed lines, shown in Supplementary Fig 5) and BCOR (solid lines, shown in Supplementary Fig 7) for original S2S reforecasts (blue) and after DL-correction (red). The horizontal line represents the BCOR of 0.5 and BMSE of 1.414.

Figure B: Lim et al. (2018) Figure 1a, b. The gray horizontal lines indicate (a) BCOR=0.5 and (b) BMSE=1.414.